# From Peak to Plunge: A Multi-Database Analysis of State-Level Disparities in Hydromorphone Use in the US

**DOI:** 10.3390/pharmacy13050147

**Published:** 2025-10-13

**Authors:** Krisha S. Patel, Leana J. Pande, Kenneth L. McCall, Brian J. Piper

**Affiliations:** 1Center for Pharmacy Innovation and Outcomes, Geisinger College of Health Sciences, Danville, PA 17821, USA; kpatel27@geisinger.edu; 2Nesbitt School of Pharmacy, Wilkes University, Wilkes Barre, PA 18766, USA; 3Garnet Health Neurology, Middletown, NY 10940, USA; lpande@garnethealth.org; 4School of Pharmacy and Pharmaceutical Sciences, Binghamton University, Johnson City, NY 13790, USA; kmccall@binghamton.edu; 5Geisinger Commonwealth School of Medicine, Department of Medical Education, Scranton, PA 18509, USA

**Keywords:** hydromorphone, opioid, Medicaid, Medicare, pharmacoepidemiology

## Abstract

Background: Hydromorphone is a semi-synthetic opioid agonist and a hydrogenated ketone of morphine. This study examined hydromorphone use in the United States (US) using three databases. Methods: The distribution of hydromorphone in the US (in grams) was provided by the US Drug Enforcement Administration’s Automated Reports and Consolidated Orders System (ARCOS) by state, zip code, and business type (pharmacies, hospitals, providers, etc.). Hydromorphone prescription claims were also examined using the Medicaid and Medicare Part D programs from 2010 to 2023. Results: Hydromorphone increased by +30.6% by 2013, followed by a decrease of −55.9% by 2023 in ARCOS. Medicaid prescriptions increased by +39.6% by 2015 and decreased by −48.9% by 2023. Medicare Part D claims increased by +8.5% by 2015 and decreased by −31.9% by 2023. There were also pronounced regional disparities in hydromorphone use identified in ARCOS (158.7-fold), Medicaid (17.5-fold), and Medicare Part D (13.7-fold). Conclusions: Hydromorphone use in the US has decreased substantially from 2010 to 2023. Additionally, these findings highlight considerable regional disparities, which may inform targeted opioid stewardship initiatives and guide policymakers to ensure safe and equitable opioid prescribing practices.

## 1. Introduction

Hydromorphone is a potent opioid approved for managing moderate-to-severe acute and severe chronic pain in patients [1], Table 1. This agent has a 5–10-fold greater analgesic effect than morphine and crosses the blood–brain barrier more easily [2]. Hydromorphone is available orally in powder, solution, immediate-release tablets, and modified-release tablets and parenterally by intravenous, intramuscular, and subcutaneous routes. It is absorbed in the upper small intestine, is extensively metabolized by the liver, and has a variety of renally excreted, water-soluble metabolites. Like morphine, it is a μ opioid receptor agonist and binds to a lesser degree to the delta receptors. Because it is more fat-soluble than morphine, its onset of action is correspondingly faster than that of morphine but is slower than highly lipid-soluble drugs such as fentanyl [3].

Hydromorphone is prescribed only when initial treatments have proven ineffective, primarily due to the drug’s elevated potency, potential for abuse, and risk of overdose [1]. It is a controlled (Schedule II) substance in the US. Acute overdose of hydromorphone can produce severe respiratory depression, somnolence progressing to stupor or coma, skeletal muscle flaccidity, cold and clammy skin, constricted pupils, reduction in blood pressure and heart rate, and death [4]. The Researched Abuse, Diversion and Addiction-Related Surveillance (RADARS) System StreetRx Program is based on crowdsourcing street prices of drugs. Site visitors spontaneously and anonymously submit the street prices they paid, or heard were paid, for diverted prescription drugs. As of 2023, based on the average prices mentioned on StreetRx (corrected by the annual rate of inflation), if the entire prescribed amount were diverted, hydromorphone has a black-market value of around USD 5.32 per mg [5].

Past research has identified pronounced geographical disparities in several opioids, including morphine, oxycodone, and codeine [6], but hydromorphone has not yet been studied. Hydromorphone warrants separate evaluation because it occupies a distinct clinical niche relative to morphine, oxycodone, and codeine. It is substantially more potent, commonly used in acute care (including parenteral formulations in hospitals and emergency departments), and often selected when first-line therapies fail or are contraindicated. As a result, hydromorphone’s temporal patterns and state-level disparities may not mirror those of more widely prescribed opioids.

Buprenorphine is an opioid with rising popularity in recent years, with patients who have opioid use disorder and depend on this agent [7]. Hydromorphone may see increasing use for acute pain treatment alongside the use of buprenorphine. Buprenorphine has a high mu receptor affinity, which can limit weak mu agonists, but hydromorphone has a stronger agonist action and can be utilized accordingly [8,9].

Prior research at Geisinger has identified a 6-fold state-level difference in oxycodone in 2021 [10] but 37-fold for oxymorphone, 30-fold for methadone, 22-fold for meperidine, 13-fold for hydrocodone, 12-fold for tapentadol, 10-fold for codeine, and 3-fold for fentanyl as well as morphine in 2019 [11]. Our multi-database approach (the Drug Enforcement Administration’s Automated Reports and Consolidated Orders System (ARCOS), distribution to retail and hospital outlets [12], Medicaid and Medicare outpatient claims [13,14]) allows us to triangulate trends across care settings and payers, identify state-level disparities, and distinguish distribution from patient dispensing. This study also examined correlations between these three datasets to better understand trends and relationships in hydromorphone distribution and prescriptions.

**Table 1 pharmacy-13-00147-t001:** Pharmacological properties of hydromorphone.

Properties	Data	Source
FDA approved for (age)	Severe pain (adults)	[15]
Pharmacodynamics	Opioid (µ agonist)	[16]
Formulations	Immediate-release and extended-release tablets and capsules, oral solution, IV	[4]
US Schedule (UK Class)	II (A)	[4]
Dose (mg/day)	2–8 ^I^, 8–32 ^E^	[4]
Half-life (hours)	8–15	[16]
StreetRx.com cost/year (2023)	USD 15,534.4 ^I^, USD 62,137.6 ^E^	[6]

I: immediate release. E: extended release.

## 2. Materials and Methods

Data sources: The US Drug Enforcement ARCOS provided data on the distribution of hydromorphone (grams) nationally, by state, zip code, and business type (pharmacies, hospitals, providers, etc.) [12]. The US Census Bureau provided data on the adults per state for 2013 and 2023 to correct the distribution for population [17].

Medicaid.gov provided the number of prescriptions (brand and generic) per state in the US from 2013 to 2023 [13]. Data.Medicaid.gov provided the number of Medicaid enrollees by state for December of 2015 and 2023 [18].

The US Centers for Medicare and Medicaid Services (CMS) provided the number of Medicare Part D claims nationally and per state for brand and generic formulations for 2013 to 2023 [14]. CMS also provided the number of Medicare enrollees by state, in December of 2015 and 2023, to correct the prescriptions for the enrollees [19]. The US Physician Workforce Data Dashboard provided the number of physicians by state in the US in 2023 [20]. The US Census Bureau provided information on the demographics of the population by state distributed by race (Whites/non-White) in the US [21].

Statistical analysis: We used Datawrapper to create heatmaps [22], GraphPad Prism (Version 10.5.0, GraphPad Software, San Diego, CA, USA) [23] to create waterfall graphs and scatterplots, and Microsoft Excel to calculate z-scores. A 95% confidence interval (mean + 1.96 × SD) was determined, and the states that fell outside this range on the waterfall plots were considered statistically (*p* < 0.05) different [24]. States that fell outside an 86% confidence interval (mean + 1.5 × SD) and the fold difference between the highest and lowest states were noted. Exploratory associations between hydromorphone use and state characteristics [20,21] were determined with Pearson r. An R value of ±0.1 to ±0.3 accounts for a small, ±0.4 to ±0.6 ≅ medium, ±0.7 to ±0.9 ≅ large, and ±0.9 to ±1.0 ≅ very large portion of the variance [25]. As states can differ in their Medicaid eligibility, exploratory analyses were also completed with prescriptions per adult population.

## 3. Results

### 3.1. ARCOS 2010–2023

There was an increase in distribution from 2010 to 2013 (+30.6%), peaking at 1839.45 kg. This was followed by a large decrease (−55.9%) by 2023, reaching a low of 810.94 kg. Overall, pharmacies accounted for the preponderance (79.8%), whereas hospitals (19.4%) and practitioners (0.8%) were responsible for a subset of distribution in 2023 (Figure 1).

The milligrams per person decreased in all but two states (NY and MS). Also, the Midwestern states (WI, ND) had a higher percentage decrease (Figure 2a). New York had the highest increase (+104.0%) while Wisconsin had the largest decrease (−74.6%). There was a greater decrease (−84.2 to −71.8%) in the smaller cities, whereas the least change (−47.0 to −2.0%) was in the bigger cities of New York. In Wisconsin, the metropolitan areas showed the largest percent decrease (−84.2 to −71.8%), whereas the rural areas showed a lower percent change (−47.0 to −2.0%) (Appendix A).

In 2023, Vermont had the highest milligrams per person (47.6, *p* < 0.05), whereas Alabama had the lowest (0.3). There was a 158.7-fold difference between them (Figure 2b).

### 3.2. Medicaid 2013–2023

From 2013 to 2015, there was an increase (+39.6%) in total prescriptions, peaking at 1192,339. There was a large decrease (−48.9%) in prescriptions by 2023, reaching a low of 609,530. The brand name Exalgo had practically vanished by 2023 (accounting for 0.0% of the total), whereas Dilaudid accounted for almost one-fifth of prescriptions (19.3%). Overall, generic (>80%) stayed consistent and made up most prescriptions (Figure 3).

By 2023, the Medicaid prescriptions decreased by a wide range in almost all states. Only a few states (MS, MA, ID, NE) showed a positive change. The Western states (WY, UT, CO) showed the most negative change, whereas the Midwestern states (IA, KS, SD) only had a small decrease (Figure 4a).

In 2023, North Dakota had the highest prescriptions per enrollee (3.5), whereas Arkansas had the lowest (0.2), showing a 17.5-fold difference between them (Figure 4b).

There was a 37.3-fold difference between the highest (Delaware = 14.9) and lowest (South Carolina = 0.4) prescribing states when correcting for adult population (Appendix A). There was a high correlation (r (49) = +0.832, *p* < 0.0001) between prescriptions corrected for enrollees and prescriptions corrected for all adults.

**Figure 3 pharmacy-13-00147-f003:**
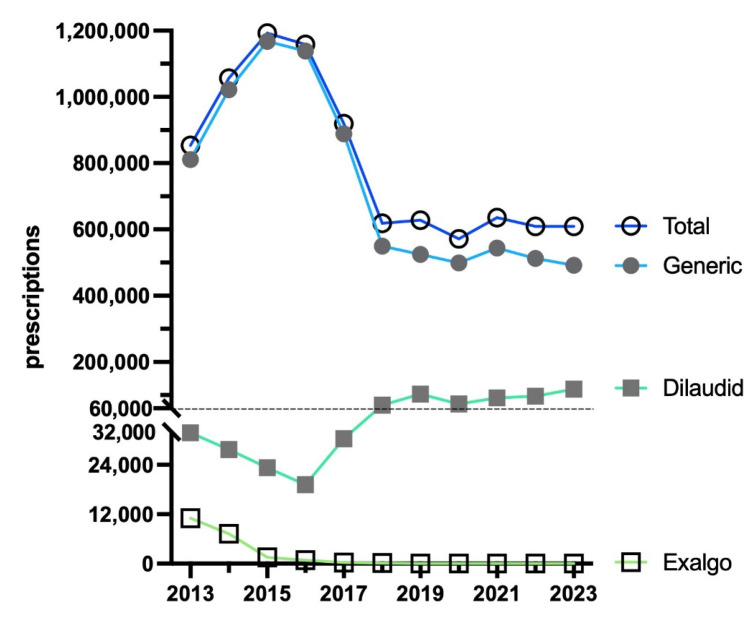
Prescriptions of hydromorphone to US Medicaid patients [9] by formulation from 2013 to 2023.

**Figure 4 pharmacy-13-00147-f004:**
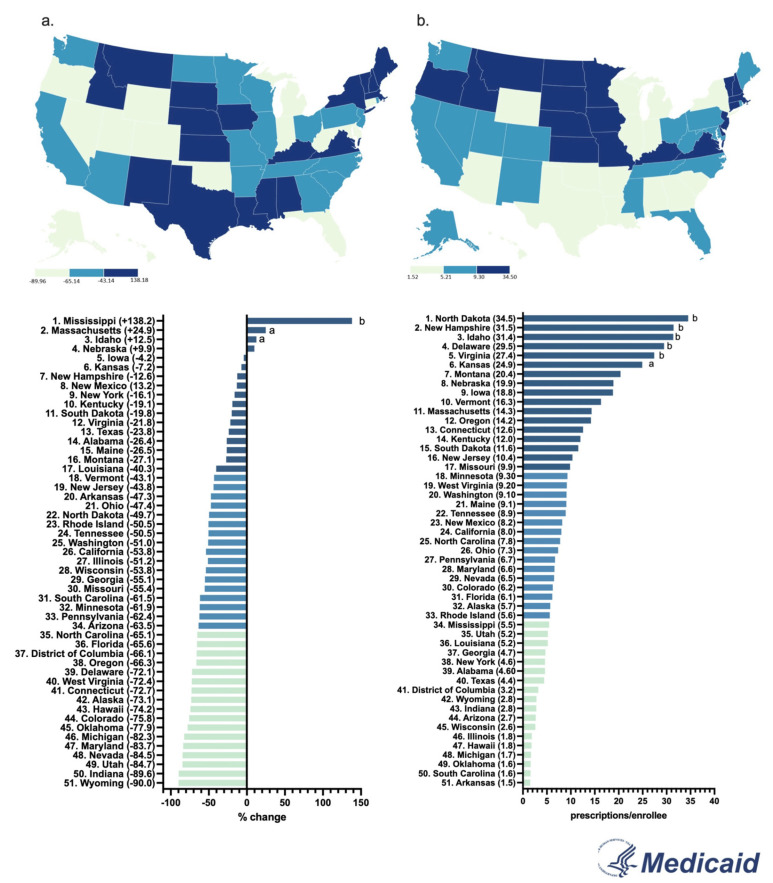
Percent change in US Medicaid prescriptions [9] per thousand enrollees from 2015 to 2023 (**a**) and prescriptions per thousand enrollees in 2023 (**b**). (States outside of ^a^1.5 or ^b^1.96 SDs).

### 3.3. Medicare 2013–2023

The data showcased an increase (+8.5%) in total Medicare claims by 2015, peaking at 1,308,316. By 2023, there was a large decrease (−31.9%) in claims, reaching a low of 891,538. The brand names Dilaudid (0.2%) and Exalgo (0.0%) had almost vanished by 2023, whereas the generic (99.8%) stayed consistent in accounting for the vast preponderance of the total Medicare claims (Figure 5).

By 2023, the claims increased by a wide range (+1.6 to 119.5%) in almost all states. A subset of states (21) showed a negative change. The Northeastern states (MN, CT, RI) increased the most, whereas the Midwestern states (ND, WV, KS) decreased the most (Figure 6a).

In 2023, Connecticut had the highest claims per enrollee (13.7), whereas West Virginia had the lowest (1.0), showing a 13.7-fold difference between them (Figure 6b).

**Figure 5 pharmacy-13-00147-f005:**
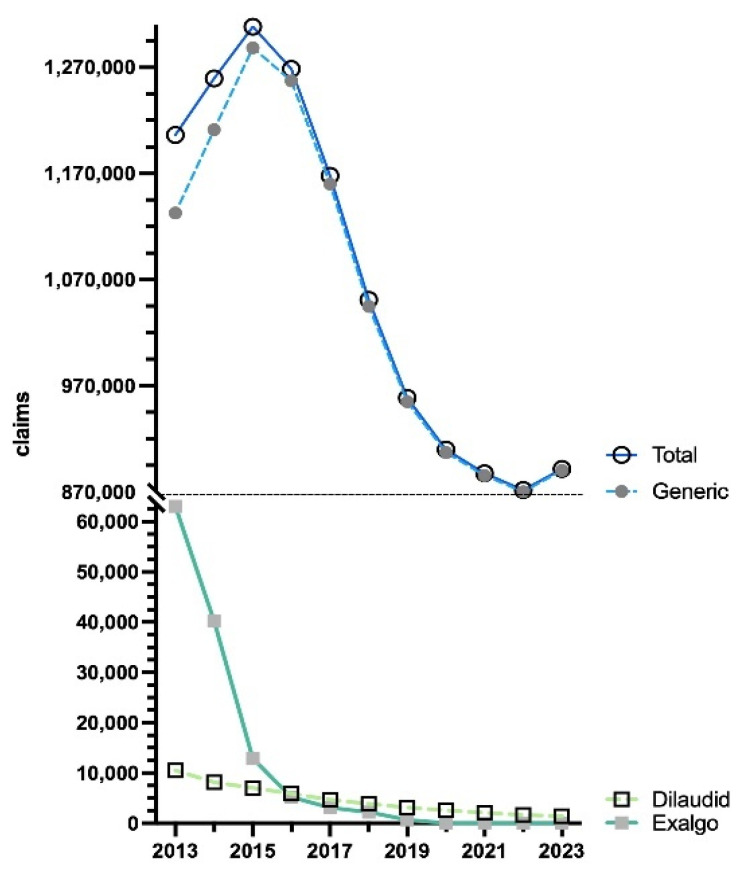
US Medicare Part D [10] claims of hydromorphone by formulation from 2013 to 2023.

**Figure 6 pharmacy-13-00147-f006:**
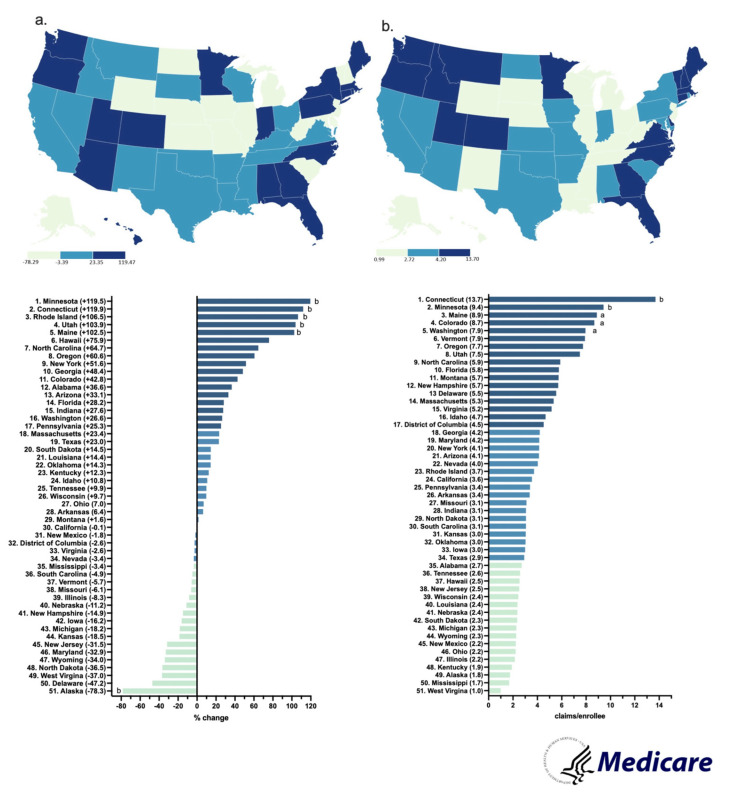
Percent change in US Medicare Part D claims [10] per enrollee from 2015 to 2023 (**a**) and claims per hundred enrollees of hydromorphone in 2023 (**b**). (States outside ^a^1.50 and ^b^1.96 SDs).

### 3.4. Correlations

ARCOS in 2013 highly correlated with ARCOS in 2023 (R^2^(49) = 0.970, *p* < 0.0001) (Figure 7a). Within Medicaid, the prescriptions in 2015 were highly associated with 2023 prescriptions (R^2^(49) = 0.422, *p* < 0.0001) (Figure 7b). In 2015, the Medicare claims were highly correlated to Medicaid prescriptions (R^2^(49) = 0.466, *p* < 0.0001) (Figure 7c). In 2023, the Medicaid prescriptions had a moderate negative correlation with the population demographics (% non-Whites) (R^2^(49) = 0.128, *p* = 0.010) (Figure 7d), and the ARCOS distribution moderately correlated with the median ages (R^2^(49) = 0.107, *p* = 0.019) (Figure 7e). Also, in 2023, the Medicare claims had a small, but significant, association with providers per thousand population (R^2^(49) = 0.085, *p* = 0.038) (Figure 7f). All other correlations between the ARCOS distribution, Medicaid prescriptions, Medicare claims, and state demographics are presented in Table 2.

## 4. Discussion

This novel study identified dynamic changes and regionally dependent disparities in hydromorphone use in the US between 2010 and 2023. According to ARCOS, hydromorphone distribution increased by +30.6% through 2013, followed by a decrease of −55.9% by 2023 (Figure 1). A similar pattern was observed in both Medicaid and Medicare programs. Medicaid hydromorphone prescriptions increased by +39.6% by 2015 and then decreased by −48.9% by 2023 (Figure 3). Similarly, Medicare Part D hydromorphone claims increased by +8.5% by 2015 and then decreased by −31.9% by 2023 (Figure 5). Given that hydromorphone is typically prescribed only when initial treatments have proven ineffective, primarily due to the drug’s elevated potency, potential for abuse, and risk of overdose [1], there is a critical need to better understand this usage pattern.

Since 2000, there has been a more than 1000% increase in opioid overdose deaths [26]. In 2021, an estimated 2.5 million people aged 18 or older had opioid use disorder, yet only about one in five received treatment [27]. By December 2023, there were 20% fewer deaths than there were in December 2022 [28]. Overall, while opioid-related deaths were declining, the ongoing opioid crisis is far from over. In 2023, nearly 110,037 people in the United States died from drug overdoses, with approximately 83,140 deaths involving an opioid [29]. A recent study among 29 states and the District of Columbia showed that the percentage of overdose deaths involving counterfeit pills more than doubled from July 2019 to December 2021. Since then, many policy interventions have targeted the opioid overdose epidemic [29]. The Overdose Data to Action initiative (established in 2019) helps local and state agencies carry out prevention strategies and collect accurate and reliable data on nonfatal and fatal overdoses [30]. The Opioid Rapid Response Program (updated in 2024) is a national initiative that aims to lower opioid overdose deaths. The program supports continuity of care and reduces risks for patients by notifying state health agencies about federal law enforcement actions that might impact a patient’s access to care [31]. In addition to these policies, states have adopted Prescription Drug Monitoring Programs (PDMPs), which are electronic databases that track controlled substance prescriptions [32]. Information from PDMPs can help clinicians identify patients who may be at risk for overdose and provide potentially lifesaving information and interventions [33]. Lastly, the Centers for Medicare and Medicaid Services implemented new coverage policies that now ensure that starting 1 January 2020, Medicare covers methadone for MAT and related services furnished by opioid treatment programs (OTPs) [34]. The decline in hydromorphone use coincides with the implementation of these policies, which may have contributed to reduced access to this opioid as part of broader efforts to address the opioid crisis in the US. However, other factors such as prescriber behavior changes, availability of alternatives, or supply chain issues could also have influenced this pattern [35].

Additionally, it is important to note that the hydromorphone aggregate production quota decreased by −71.5% from 2015 to 2023. The DEA’s quota is adjusted “in order to provide for the estimated medical, scientific, research, and industrial needs of the U.S., lawful export requirements, and the establishment and maintenance of reserve stocks” [36]. The timing of the DEA’s aggregate production quota decrease suggests it may have played a role in the decline of hydromorphone distribution from 2010 to 2023. However, this association should be interpreted cautiously given the potential influence of other concurrent policy and market changes. Importantly, hydromorphone was distributed primarily to pharmacies (79.8%), with hospitals (19.4%) and practitioners (0.8) being a small subset of the total distribution in 2023 (Figure 1). Nearly 7000 pharmacies have closed since 2019. Many community pharmacies are struggling to stay open due to limited workers and shrinking reimbursement rates for prescription drugs [37]. The closure of pharmacies also contributes to the decline in hydromorphone distribution shown in the ARCOS data.

By 2023, New York State had an increase in hydromorphone distribution of +104.0%, with the elevation occurring largely in the state’s major metropolitan areas. However, Wisconsin had the highest decrease of −74.6%, with the reduction also concentrated primarily in the metropolitan areas of the state (Appendix A). These disparities between states showcase that there might be something more than just the pharmacy closures driving the decline of hydromorphone. The population of New York is increasing every day, reaching a high of 15,611,308 adults in 2023 [17]. Notably, since the data is corrected by population with milligrams of hydromorphone per adult in New York, the increase in the population drives the elevation in New York’s distribution as well as the regional increase in the larger, more populated, cities in the Empire State. Additionally, there were pronounced regional disparities in hydromorphone use identified in ARCOS (158.7-fold), Medicaid (17.5-fold), and Medicare Part D (13.7-fold) (Figure 2b, Figure 4b and Figure 6b). Medicaid prescriptions per thousand enrollees of hydromorphone had decreased in all but four states (MS, MA, ID, NE) from 2015 to 2023. Regionally, the Western states like Wyoming, Utah, and Colorado had the most negative change, whereas the Midwestern states like Iowa, Kansas, and South Dakota only had a small negative change (Figure 4a). Regionally, the Northeastern states like Minnesota, Connecticut, and Rhode Island increased the most, whereas Midwestern states like North Dakota, West Virginia, and Kansas decreased the most (Figure 6a). However, future studies can look into disparities at a state level over time.

Even though the overall Medicare Part D claims for hydromorphone decreased nationally, the per capita (claims/thousand enrollee) for each state actually increased by a wide range in most states from 2015 to 2023. According to the CMC’s Medicare Part D prescribing guide, all Part D plans have a Drug Management Program that limits access to opioids that are frequently abused. This includes the seven-day supply limit alert that “limits initial opioid fills for Part D patients who haven’t filled an opioid prescription recently, such as within the past 60 days, to a supply of 7 days or less. This alert shouldn’t affect patients who already take opioids” [38]. This suggests that access to opioids is tightened for new patients, but the original patients who already receive prescriptions for opioids are not affected. For example, some states might have prescribed hydromorphone to fewer patients, but those who did might have received the drug in more concentrated and more frequent prescriptions. This could explain why the hydromorphone Medicare claims in the US decreased overall from 2015 to 2023, but the per capita claims increased in most states.

These three databases were correlated with each other and state population demographics. The 2013 ARCOS was very highly correlated (r = 0.984) with the 2023 ARCOS distribution (Figure 7a). This indicated a long-term consistency in the distribution of hydromorphone. Similarly, there was a moderate correlation (r = 0.649) between the 2015 Medicaid and the 2023 Medicaid prescriptions (Figure 7b). This indicated persistent state-level differences in the Medicaid prescriptions over time. There was also a high correlation (r = 0.683) between the Medicaid 2015 prescriptions and Medicare 2023 claims (Figure 7c). This indicated some level of shared prescribing environment or policies between the two public healthcare programs as well.

Additionally, there was a negative moderate association (r = −0.357) between the Medicaid prescriptions and the non-White population per state in 2023 (Figure 7d). This finding may reflect potential racial disparities in access to hydromorphone, as states with higher non-White populations tended to have lower Medicaid prescriptions. CDC data indicate that treatment access is lower among Black, Hispanic, and American Indian or Alaska Native populations [39], which may partially explain this pattern. Nonetheless, differences in disease prevalence, clinical guidelines, or prescriber practices across states could also contribute. There was also a low correlation (r = 0.328) between the ARCOS distribution and the average median age per state in 2023 (Figure 7e). This suggested the age-related needs of hydromorphone in states where the median age was higher. The CDC stated in a recent data brief that in 2023, 24.3% of adults had chronic pain, and 8.5% of adults had chronic pain that also increases with age [40]. This would account for why hydromorphone distribution is higher in states with higher median ages. Lastly, there was a low correlation (r = 0.292) between the Medicare 2023 claims and the providers per thousand population per state in 2023. This suggested an access-related disparity where states with a lower number of providers had fewer Medicare claims. The Association of American Medical Colleges predicted that the United States will face a physician shortage of up to 86,000 physicians by 2036 [41]. The Kaiser Family Foundation also stated that nearly three million Americans live in rural areas that lack both healthcare and internet access, which in turn creates access barriers for patients [42]. This also provides a possible explanation as to why there were fewer claims in the states with a lower number of providers.

These persistent and significant state-level disparities suggest that regional prescribing cultures and healthcare patterns continue to influence the geographic variation in opioid prescribing. These findings stress that for opioid policies, there is no one-size-fits-all solution, and that many states and regions of the US are influenced by other outcomes like access, population, and demographics as well. The clinical utility of hydromorphone is inherently limited by its pharmacological profile. While its high lipophilicity and rapid onset make it effective for severe pain management, these same properties contribute to its high abuse potential and overdose risk. The drug’s primary FDA indication for use, “for the management of pain severe enough to require an opioid analgesic and for which alternate treatments are inadequate only when other treatments have failed”, reflects appropriate clinical caution. However, the historical prescribing patterns and state-level variation revealed in the study suggest this restriction may not have been consistently applied. The availability of multiple formulations (oral, parenteral) and the drug’s ability to cross the blood–brain barrier more readily than morphine underscore its utility for pain management and the need for careful patient selection and monitoring [3,43,44].

Several state-level disparities were shown through this study, with a decline in hydromorphone use being the primary finding. Disparities that include varying levels of data availability, opioid use, and mortality rates were prevalent across the different states. This complexity is compounded by a rise in chronic pain conditions, underemphasized education for healthcare providers in pain management, and a limited number of specialists in pain management and addiction medicine. To combat these disparities, the American Hospital Association has implemented opioid stewardship programs that promote appropriate use of opioid medications, improve patient outcomes, and reduce misuse of opioids [45]. These programs will continue to focus on prescription opioids in the future to reduce state-level disparities and declines that have been observed through this study. In addition to professional public health programs, we also cannot discount the contribution that patients’ attitudes towards prescription opioid use may have had on the overall declines. Concerns about misuse may have become more prominent over the past two decades [46]. The protagonists of popular shows like House (running from 2004 to 2012) [47] and Nurse Jackie (2009 to 2015) [48] both struggled with prescription opioid addiction and could have influenced patients’ views on chronic prescription opioid use. However, there are many other factors influencing patient views on opioids besides pop culture references and media, such as the highly relevant and ubiquitous nature of opioid use in the US, or even policy changes. Thinking more broadly, policies that provide universal guidelines [29,38] for opioid use and pain seek to provide a simplified solution to a multifaceted problem. Policy responses to the opioid epidemic, such as declines in production quotas [36], may help to reduce prescription opioid misuse, but increase heroin use over time [49]. Moreover, the source of opioids for illicit drug use is not necessarily from prescriptions. Drug trafficking organizations increasingly use fentanyl for heroin and counterfeit pills stamped to look like prescription drugs [50]. While broad measures like limiting the supply of opioids have positive public health effects, they can decrease the quality of individual pain management for patients with legitimate pain concerns [51,52]. Regional trends in the use of opioids may be the result of bias in gender or race that is already well-documented in pain medicine [52,53,54], but may be reinforced by external pressures to decrease the use of medications like hydromorphone. Female patients in emergency departments are less likely to be prescribed pain relief medications compared to males, even after adjusting for reported pain scores and numerous patient, physician, and ED variables [54]. Pain is highly individualistic, and people experiencing pain have different tolerances for pain and responsiveness to medications available [53,54]. Unrelieved pain is the most common motive for prescription opioid misuse among older adults (84.7%) and adolescents (56.1%) [55,56]. The majority of US adults aged 18 and older (63.4%) noted prescription opioid misuse with untreated pain as their single most important motive, and limited data exists to quantify the associations in risk of unrelieved pain and risk for OUD, suggesting monitoring patients’ pain levels and progress might be more helpful in preventing misuse than policies [52,57]. Additional potential drivers of the decline may be changes in prescribing behavior, drug shortages, shifts to alternative opioids, or evolving pain management practices. Therefore, more research needs to identify relationships between specific treatments in pain management alongside patients for more effective means of deterring the misuse of opioids.

The heavy economic burden of opioid abuse has led this to be a priority in the United States [57]. A study in Germany found that a history of pain or musculoskeletal and connective tissue disorders was associated with over 30% of reports of abuse. Depression was reported in over 10% of reports of abuse. Hydromorphone accounted for 2% of the reports of abuse [58]. However, abuse is a difficult topic to study, given limitations in demographics associated with reported cases of abuse, such as sex and age, including underreporting in totality [58]. Underreporting and overall a lack of awareness in the reporting system are substantial gaps in our knowledge of abuse [56]. Stigma regarding these medications can limit reports of adverse drug reactions, as can fear of loss of future access to prescriptions [55].

Opioids are high-risk pharmacotherapies for medication errors, and unfortunately, they are a necessity for managing pain [59]. Chronic and severe pain patients often have to depend on opioids due to inadequate existing alternatives [55]. Palliative and hospice care patients can primarily depend on opioids to manage pain and other symptoms like dyspnea and cough [59]. A qualitative study found that patients suffered severe overdosage and withdrawal symptoms from inappropriate dosing after opioid switching [59]. In home settings with non-professional caregivers, an administration error incidence of 49% was reported in a longitudinal study conducted over three consecutive days [59]. Opioids have the highest rates of medication error amongst the medications prescribed in oncology [59]. This can in part be explained by the complexity of conversion between different opioid formulations as patients’ needs change [59]. Additionally, there is controversy over the topic of what qualifies as abuse or medication errors [60]. A medication error is a preventable event that may lead to inappropriate medication use or harm to the patient; misuse or abuse is taking medication in a way that is not prescribed. Misuse or abuse may also not be the result of addiction, but uncontrolled pain. Inadequate prescription practices can bring patients to situations of abuse or misuse. Medication errors can exist in both over- and under-prescribing [60].

A limitation that comes with the comprehensive ARCOS database is that it does not provide any information on how many people are receiving hydromorphone or who is receiving it; it simply provides the weight of the drug distributed to each state. However, Medicaid and Medicare Part D programs fill that gap by providing information on the number of prescriptions and claims for hydromorphone. This extends previous reports that looked at only distribution [6]. ARCOS reporting by manufacturers and distributors of Schedule II drugs such as hydromorphone is federally mandated and audited, and coverage is generally considered complete. Prior work has shown a very high correlation between ARCOS opioid volumes and commercial distribution/dispensing datasets, supporting its reliability for temporal analyses [35]. Medicaid State Drug Utilization Data includes outpatient pharmacy claims at the NDC–state–quarter level and is widely used, but it has known limitations, including incomplete managed care reporting historically. Medicare Part D may have NDC-level coding issues and variation in days’ supply, but the overall coverage and internal consistency are high. We mitigated these known limitations by reviewing the dataset for any obvious anomalies. Any residual bias is most likely toward underestimation of absolute volumes rather than distortion of relative patterns. The correlations between the three databases suggest that the lack of specificity in ARCOS was in fact captured by Medicaid and Medicare. However, the Medicaid and Medicare databases do not contain additional variables with the demographic or other clinical drivers of disparities. Another limitation of the study is the absence of clinical data at the patient level. For example, factors like the inability to assess the appropriateness of prescribing or patient outcomes may also be contributing to the regional disparities. Therefore, future studies can focus on looking at individual patient data to understand the disparities at the patient level in addition to the ecological-level disparities found in this study. For the future, it is important that interventions and policies are regionally tailored to account for the demographic and geographical differences that impact access not only to medications, but also to healthcare in general.

## 5. Conclusions

This multi-database analysis demonstrates that hydromorphone utilization in the United States has undergone a significant transformation over the past decade, with a marked decline following peak usage around 2015. The convergent findings across ARCOS, Medicaid, and Medicare data sources provide robust evidence that policy interventions and clinical guideline changes may have successfully reduced access to this high-potency opioid. However, persistent regional disparities highlight the need for continued surveillance and targeted interventions to ensure appropriate pain management while minimizing opioid-related harms.

## Figures and Tables

**Figure 1 pharmacy-13-00147-f001:**
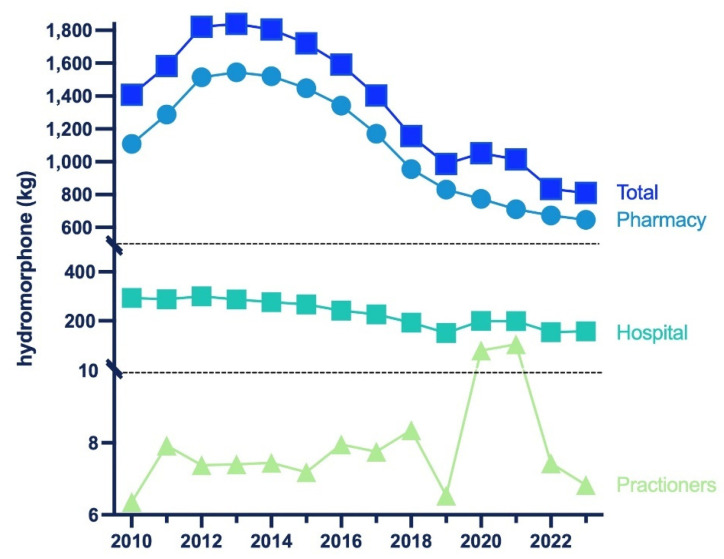
Hydromorphone in kilograms distributed by business activity from 2010 to 2023 reported by the US Drug Enforcement Administration’s Automated Reports and Consolidated Orders System (ARCOS).

**Figure 2 pharmacy-13-00147-f002:**
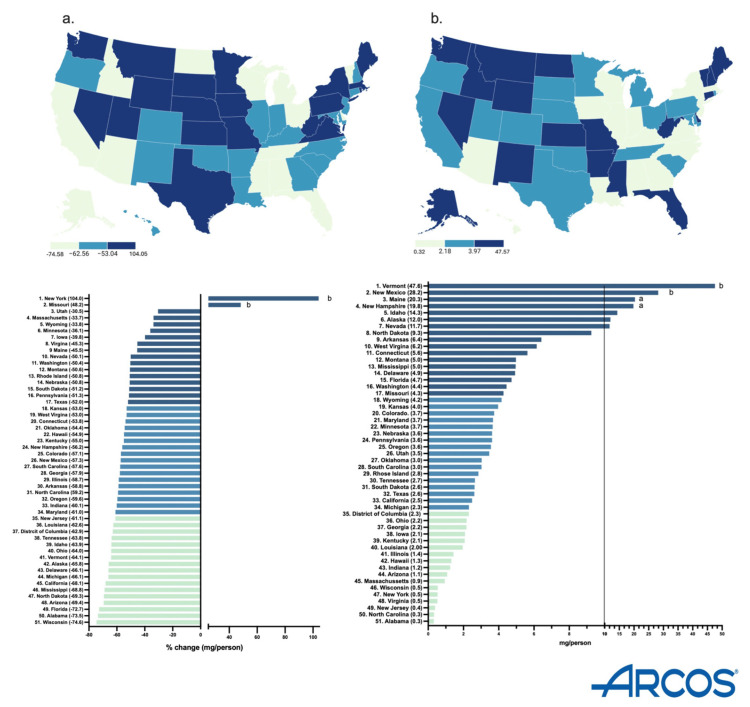
Drug Enforcement Administration’s Automated Reports and Consolidated Orders System (ARCOS) percent change in mg/person of hydromorphone from 2013 to 2023 (**a**) and mg/person in 2023 (**b**). (States outside of ^a^1.5 or ^b^1.96 SDs).

**Figure 7 pharmacy-13-00147-f007:**
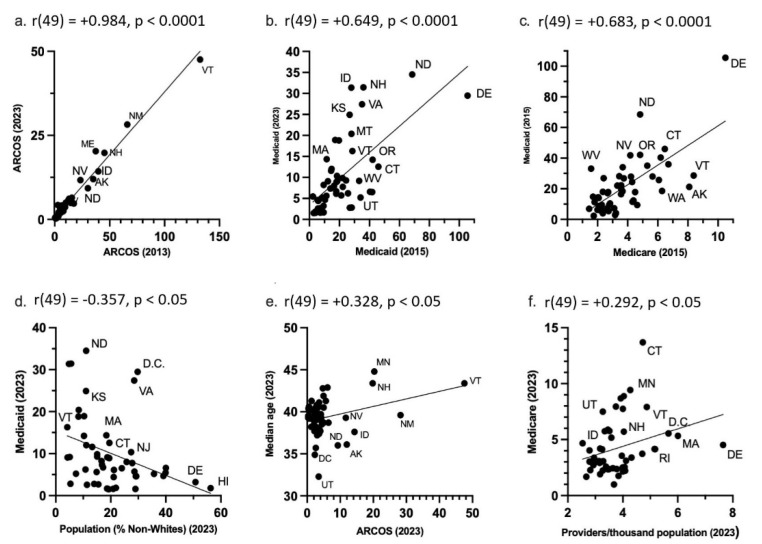
State-level ARCOS, Medicaid, and Medicare scatterplots. (**a**) ARCOS 2013 mg/person correlation with ARCOS 2023 mg/person. (**b**) Medicaid 2015 prescriptions/thousand enrollees correlated with Medicaid 2023 prescriptions/thousand enrollees. (**c**) Medicare 2015 claims/hundred enrollees correlated with Medicaid 2015 prescriptions/thousand enrollees. (**d**) Population percentage of non-Whites in 2023 correlated with Medicaid 2023 prescriptions/thousand enrollees. (**e**) ARCOS 2023 mg/person correlated with 2023 population median age. (**f**) Providers/thousand population in 2023 correlated with Medicare 2023 claims/hundred enrollees. The dots represent datapoints, the line represents the line of best fit, and the abbreviations are for the U.S. states.

**Table 2 pharmacy-13-00147-t002:** ARCOS, ARCOS, Medicaid, and Medicare correlations with state-level demographics including median age [14], percent non-White [18], providers per thousand population [17] (* *p* < 0.05, ** *p* < 0.005, and *** *p* < 0.0001).

	Medicaid (2015)	Medicare (2015)	ARCOS (2023)	Medicaid (2023)	Medicare (2023)	Median Age (2023)	% Non-Whites (2023)	Providers/Thousand People (2023)
ARCOS (2013)	0.124	0.408 **	0.984 ***	0.266	0.199	0.293 *	−0.277 *	0.056
	Medicaid (2015)	0.683 ***	0.117	0.649 ***	0.337 *	−0.247	−0.175	0.152
		Medicare (2015)	0.389 **	0.488 **	0.572 ***	−0.041	−0.073	0.385 *
			ARCOS (2023)	0.258	0.233	0.328 *	0.325 *	0.042
				Medicaid (2023)	0.192	0.098	0.357 *	0.015
					Medicare (2023)	0.162	0.195	0.292 *
						Median age (2023)	−0.049	0.299 *
							% Non-Whites (2023)	0.320 *

## Data Availability

Raw data is available [12,13,14]. Extracted data From Peak to Plunge: A Multi-Database Analysis of State Level Disparities in Hydromorphone use in the U.S. (2010–2023)|medRxiv.

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
