# Peer review of "From Peak to Plunge: A Multi-Database Analysis of State-Level Disparities in Hydromorphone Use in the US"

_pharmacy, 2025, doi:10.3390/pharmacy13050147_

Round 1

Reviewer 1 Report

Comments and Suggestions for Authors

Dear authors

Congratulations on your manuscript, “From Peak to Plunge: A Multi-Database Analysis of State Level Disparities in Hydromorphone Use in the U.S.”, on a dated and relevant topic. To improve your manuscript, I share some questions/suggestions as follows.

Areas for Improvement

  1. Causality vs. Correlation
  • Topic Several discussion points imply causal relationships between policy changes (e.g., DEA quotas, PDMPs, Medicare restrictions) and the decline in hydromorphone use, although the study is observational and not designed to establish causation.
  • Suggestion Try to soften causal language; emphasise temporal associations and acknowledge other possible influencing factors (prescriber behaviour, drug shortages, alternative therapies, population health trends).

  1. Limited Patient-Level Context
  • Topic The study relies on state-level, aggregate data. This means individual-level prescribing patterns, patient diagnoses, and clinical appropriateness cannot be assessed.
  • Suggestion Explicitly note this as a limitation and, if possible, suggest future linkage with patient-level or prescription-level datasets to better understand clinical drivers of disparities.

  1. Racial Disparity Interpretation
  • Topic The correlation between % non-White populations and lower Medicaid prescriptions is interpreted as an access disparity, but other factors (e.g., differences in pain prevalence, prescribing culture, state Medicaid coverage rules) could contribute.
  • Suggestion Present this as a potential disparity, discuss possible confounders, and recommend targeted studies to clarify causes.

  1. Figures and Tables
  • Topic Figures are informative but not always fully self-explanatory without cross-referencing text.
  • Suggestion Strengthen figure legends to include brief methodological notes (e.g., how “fold-differences” are calculated, definition of “per enrollee” metrics) so they can stand alone.
  1. Policy Impact Interpretation
  • Topic The discussion attributes observed trends to policy measures without fully exploring alternative explanations such as market supply disruptions, shifts to other opioids, or broader changes in pain management practices.
  • Suggestion Add a short section contrasting the observed hydromorphone trends with trends in other opioids to contextualize whether declines are drug-specific or part of a broader opioid reduction pattern.

  1. Literature Context
  • Topic While references are appropriate, the study could be strengthened by more explicitly comparing its findings with previous opioid disparity studies.
  • Suggestion Add 2–3 sentences in the discussion highlighting how hydromorphone trends compare with other opioids, citing recent U.S. pharmacoepidemiology literature.

Questions for the Authors

Some questions to help the authors understand the point of view of this review.

  1. Causal Language
    • Several discussion points suggest that specific policy interventions (DEA quotas, PDMPs, Medicare restrictions) caused the decline in hydromorphone use. Could you clarify these statements to emphasise temporal association rather than direct causality?
  2. Alternative Explanations
    • Beyond policy changes, did you explore other potential drivers for the observed decline, such as changes in prescriber behaviour, drug shortages, shifts to alternative opioids, or evolving pain management practices? If not, could you briefly discuss these possibilities?
  3. Racial Disparity Findings
    • The correlation between % non-White populations and lower Medicaid prescriptions is interpreted as a disparity in access. Could you discuss potential confounding factors, such as disease prevalence differences, prescribing guidelines, or state Medicaid policies?
  4. Comparison With Other Opioids
    • Would adding a short comparison of hydromorphone trends with other opioids strengthen the context of your findings? This could help determine whether the observed pattern is unique to hydromorphone.
  5. Figures and Legends
    • Some figure legends require cross-referencing the methods to fully understand metrics (e.g., fold-difference calculations, “per enrollee” definitions). Could you expand legends to make them more self-contained?
  6. Limitations Section
    • You mention the limitation of aggregate data, but could you elaborate on how the absence of patient-level clinical details might affect interpretation (e.g., inability to assess appropriateness of prescribing or patient outcomes)?
  7. Policy Impact Scope
    • Since policy changes occurred at different times across states, did you consider a state-by-state temporal analysis to better link specific interventions with changes in prescribing?
  8. Data Source Reliability
    • Could you comment on the completeness and reliability of the ARCOS, Medicaid, and Medicare datasets, especially in terms of possible underreporting or coding inconsistencies?

Overstated claims

Please check the conclusions to soften some claims. Above you find some cases with suggestions.

  1. Decline in hydromorphone use and policy interventions

Original (lines 207–209)

“The decline of hydromorphone use suggests that these policies may have been effective in reducing access to the opioid as part of efforts to control the opioid crisis in the US.”

Suggested rewording

“The decline of hydromorphone use coincides with the implementation of these policies, which may have contributed to reduced access to the opioid as part of broader efforts to address the opioid crisis in the US. However, other factors such as prescriber behaviour changes, availability of alternatives, or supply chain issues could also have influenced this trend.”

  1. Racial disparities in Medicaid prescriptions

Original (lines 265–270)

“This indicated the racial disparity in access to the opioids, where states with higher non-White populations had lower Medicaid prescriptions. The CDC’s overdose prevention website states that the percentage receiving treatment in the US was lowest for Black, Hispanic, and American Indian or Alaska Native racial and ethnic groups [38]. This could explain the racial disparity seen in the Medicaid 2023 prescriptions.”

Suggested rewording

“This finding may reflect racial disparities in access to hydromorphone, as states with higher non-White populations tended to have lower Medicaid prescriptions. CDC data indicate that treatment access is lower among Black, Hispanic, and American Indian or Alaska Native populations [38], which may partially explain this pattern. Nonetheless, differences in disease prevalence, clinical guidelines, or prescriber practices across states could also contribute.”

  1. Impact of DEA production quotas

Original (lines 213–215)

“Our data suggests that the DEA’s aggregate production quota decrease was probably a factor in the slow decline of hydromorphone distribution from 2010 to 2023 in the US as well.”

Suggested rewording

“The timing of the DEA’s aggregate production quota decrease suggests it may have played a role in the decline of hydromorphone distribution from 2010 to 2023, although this association should be interpreted cautiously given the potential influence of other concurrent policy and market changes.”

Ethical Compliance Points

Despite no evidence of ethical misconduct or breaches in human protection, improvements would mostly be in responsible interpretation to avoid overstatement and ensure readers understand the limitations of correlation-based analyses.

Potential Minor Ethical Considerations

  1. Overinterpretation of disparities
    • While not unethical per se, attributing state-level racial differences directly to access issues without adjusting for confounders could unintentionally oversimplify complex sociopolitical issues.
    • It’s important they frame these results cautiously to avoid misrepresentation.

  1. Policy impact statements
    • The discussion sometimes implies policy measures caused observed changes. If policymakers act on overinterpreted conclusions, this could have unintended consequences. This is more about scientific responsibility than misconduct.

  1. Self-citation bias
    • Although not excessive, the relatively high visibility of the authors’ previous work in the reference list warrants transparency that these are related studies forming part of an ongoing research program.

References improvement

Regarding the references, I leave, at the author’s discretion, a set of references and their potential value to enhance the manuscript.

Most valuable suggestions for this manuscript

  • #1, #2 → to broaden the discussion on misuse, abuse, and pharmacovigilance findings.
  • #4, #5 → to add a patient safety and medication error perspective, which is currently underrepresented in the paper.
  • #3 could be optional, depending on whether the authors want to expand into treatment alternatives.

  1. Jobski et al., 2023 – EudraVigilance analysis of opioid-associated abuse, dependence, withdrawal

Jobski, K.; Bantel, C.; Hoffmann, F. Characteristics and completeness of spontaneous reports by reporter’s role in Germany: An analysis of the EudraVigilance database using the example of opioid-associated abuse, dependence, or withdrawal. Pharmacol. Res. Perspect. 2023, 11, e01077.

Value  Would add an international pharmacovigilance perspective, showing how spontaneous reporting can identify abuse and dependence signals, complementing the U.S. database analysis in this paper.

  1. Gustafsson et al., 2024 – Systematic review on opioid misuse, abuse, and medication errors

Gustafsson, M.; Silva, V.; Valeiro, C.; Joaquim, J.; van Hunsel, F.; Matos, C. Misuse, Abuse and Medication Errors’ Adverse Events Associated with Opioids—A Systematic Review. Pharmaceuticals 2024, 17, 1009. https://doi.org/10.3390/ph17081009

Value Highly relevant — synthesises global evidence on non-therapeutic opioid use and errors, strengthening the discussion on safety concerns and placing hydromorphone trends in a broader misuse/abuse context.

     3.Dennis et al., 2014 – Effectiveness of opioid substitution treatments

Dennis, B.B., Naji, L., Bawor, M. et al. The effectiveness of opioid substitution treatments for patients with opioid dependence: a systematic review and multiple treatment comparison protocol. Syst Rev 3, 105 (2014). https://doi.org/10.1186/2046-4053-3-105

Value Useful if the discussion is expanded to treatment alternatives for opioid dependence; could contextualize hydromorphone's place in therapy and harm reduction strategies.

      4. Heneka et al., 2018 – Opioid errors in palliative care

Heneka N, Shaw T, Rowett D, et al. Opioid errors in inpatient palliative care services: a retrospective review. BMJ Supportive & Palliative Care 2018;8:175-179. https://doi.org/10.1136/bmjspcare-2017-001417

Value Relevant to highlight medication safety issues in high-risk care settings, especially since hydromorphone is often used in oncology and palliative care.

      5. Heneka et al., 2015 – Systematic review on opioid medication errors in oncology/palliative care

Heneka, N., Shaw, T., Rowett, D., & Phillips, J. L. Quantifying the burden of opioid medication errors in adult oncology and palliative care settings: A systematic review. Palliative Medicine, 30(6), 2015. https://doi.org/10.1177/0269216315615002

Value Complements #5 by quantifying the problem and could strengthen the argument for stewardship programs and safe prescribing guidelines.

Reviewer 2 Report

Comments and Suggestions for Authors

Thank you for the opportunity to review From Peak to Plunge: A Multi-Database Analysis of State Level Disparities in Hydromorphone Use in the U.S. for publication. This article seeks to describe trends in hydromorphone distribution across the United States.

Major comments are below:

  1. Intro: The article would benefit from a stronger description of its purpose. What will evaluating hydromorphone specifically tell us that the previous studies evaluating morphine, oxycodone, and codeine did not? Or what will the identified trends in this study be used for?
  2. Intro: while it is not the most common opioid, hydromorphone is a top choice for acute pain management in patients with OUD who are managed on buprenorphine, given its competitive binding for the mu receptor. This is an important consideration as policy interventions are made to (hopefully) increase access to MAT and therefore, when these patients require acute pain management, hydromorphone claims may also increase compared to other opioids
  3. Intro: It is unclear what the discussion surrounding the street value of medications adds to this paper about the prescription distribution of hydromorphone. It is also unclear how to interpret the dollar amount presented. It would be more digestible for the reader if it were presented as $$ per tablet. For example, a buprenorphine/naloxone film goes for about $10/film, depending on the region.
  4. Results: Figures 1, 3, and 5 use broken y axes. A broken y-axis can be misleading because it exaggerates small differences by distorting the visual scale, making effects appear larger than they are. It also reduces transparency, as readers may not realize part of the axis is omitted and misinterpret the true magnitude of the data. For example, in Figure 1, it seems that prescribers exhibit a much larger portion of the hydromorphone distribution than the true data shows. Consider an alternative graph to depict the percentage of the total that each source (pharmacies, hospitals, practitioners) makes up.
  5. Results: The text in Figures 2, 4, and 6 is incredibly small, even when zoomed in on a computer screen. Consider increasing the figure size for readability.
  6. Discussion: The discussion could benefit from a comparison to previous studies analyzing trends in other opioids in order to determine if these trends are specific to hydromorphone or can be generalized to opioids as a class. Further, how do these regional disparities compare to regional disparities in overdose (both fatal and nonfatal)?
  7. Discussion: There is an incredible emphasis on policy interventions being the cause of the decrease in distribution since 2015, but authors do not provide dates for all of these policies or describe how they affect hydromorphone specifically.
  8. Discussion: Based on the information presented, it seems that the quotas are likely the primary cause of the decline in distribution since 2015
  9. The discussion on lines 313-316 is important to emphasize that the opioid overdose conversation is not siloed among healthcare providers. However, it is unlikely that two television shows caused this concern, but rather the highly relevant and ubiquitous nature of opioid use in the US.
